# Age-Dependent Intestinal Repair: Implications for Foals with Severe Colic

**DOI:** 10.3390/ani11123337

**Published:** 2021-11-23

**Authors:** Sara J. Erwin, Anthony T. Blikslager, Amanda L. Ziegler

**Affiliations:** College of Veterinary Medicine, North Carolina State University, Raleigh, NC 27607, USA; sjerwin@ncsu.edu (S.J.E.); anthony_blikslager@ncsu.edu (A.T.B.)

**Keywords:** horse, colic, ischemia-reperfusion injury, intestinal barrier repair, enteric nervous system, enteric glial cells, tight junction proteins

## Abstract

**Simple Summary:**

Equine colic places a substantial financial burden on horse owners and the equine industry each year. Equine veterinary research is focused on preventing colic on the farm whenever possible and improving treatment options available to veterinarians in the field and referral hospitals. It is important for scientists to have a detailed understanding of the intestinal damage created by different types of colic in foals and adult horses so they can better target certain cell types or tissue systems when investigating new treatment options. This review article summarizes recent works in the field of intestinal injury research and describes the potential roles of various intestinal systems, such as the enteric nervous system (ENS), in repairing the intestine after colic injury and how these systems mature in early life.

**Abstract:**

Colic is a leading cause of death in horses, with the most fatal form being strangulating obstruction which directly damages the intestinal barrier. Following surgical intervention, it is imperative that the intestinal barrier rapidly repairs to prevent translocation of gut bacteria and their products and ensure survival of the patient. Age-related disparities in survival have been noted in many species, including horses, humans, and pigs, with younger patients suffering poorer clinical outcomes. Maintenance and repair of the intestinal barrier is regulated by a complex mucosal microenvironment, of which the ENS, and particularly a developing network of subepithelial enteric glial cells, may be of particular importance in neonates with colic. Postnatal development of an immature enteric glial cell network is thought to be driven by the microbial colonization of the gut and therefore modulated by diet-influenced changes in bacterial populations early in life. Here, we review the current understanding of the roles of the gut microbiome, nutrition, stress, and the ENS in maturation of intestinal repair mechanisms after foaling and how this may influence age-dependent outcomes in equine colic cases.

## 1. Introduction

Colic is the leading cause of death in adult horses from 1 to 20 years of age, and the most fatal form of colic in adult horses is strangulating obstruction [1]. In foals, the leading cause of strangulating obstruction is small intestinal volvulus, while large colon volvulus is rare. Age-dependent outcomes in equine patients undergoing surgical correction of small intestinal strangulating obstruction (SISO) have not been examined directly, though reported survival rates are much poorer in foals than adults (20% compared to 20–61%, respectively) [2]. Similarly, the study of comparative large animal models, such as surgically-induced SISO in the pig, have shown neonates have a limited ability to repair ischemic intestinal damage as compared to older animals. Remarkably, neonatal repair can be induced in ex vivo models [3] and studies are ongoing to understand the underlying mechanisms of age-dependent intestinal repair. In another comparative pig model, the effects of prebiotic fiber on the early postnatal development of the intestinal microbiota and the enteric glial network are under investigation, under the hypothesis that modulating the intestinal microbial populations through dietary prebiotic supplementation may drive earlier maturation of intestinal repair mechanisms, limiting damage and improving survival following ischemic injury. It is reasonable to presume that intestinal ischemic injury would be of particular consequence in foals experiencing severe colic based on the outcomes of ischemic injury seen in other mammalian species. In a comparative example, necrotizing enterocolitis (NEC) is an important condition in human neonates, especially in premature and low-birth-weight infants, where it affects up to 7% of that population [4,5]. NEC is often accompanied by up to 30% mortality and severe long-term morbidities in up to 25% of survivors that affect quality of life, and recent studies have hypothesized that NEC is a delayed-onset response to intestinal injury in utero or during parturition [4,5]. Of relevance to the comparative pig model, the porcine epidemic diarrhea virus is characterized by inflammation-driven intestinal ischemia and leads to 50–80% mortality, with up to 100% mortality reported in nursing piglets [6].

The intestine is a complex organ responsible for absorbing water and nutrients while simultaneously preventing harmful bacteria and their toxins from entering host tissues. The intestinal barrier is formed by a single layer of epithelial cells connected by intercellular tight junction proteins and is particularly vulnerable to injury. SISO causes rapid epithelial cell damage and compromises the intestinal barrier, allowing bacteria to translocate into the systemic vasculature, causing sepsis, systemic inflammatory syndrome, and multiple organ failure [7,8]. In mature animals, the barrier repairs following injury through a series of coordinated events including epithelial restitution and tight junction closure and is regulated intensively by complex and coordinated subepithelial cell signaling events [7,8]. A dense network of neurons and glia comprising the ENS is known to regulate the epithelial barrier but, important to neonates, it is immature at birth [9]. Postnatal development of the ENS, particularly the migration of enteric glial cells into the lamina propria, is thought to be driven by microbial colonization after birth and is likely modulated by diet-influenced changes in microbial populations in early life [3,9]. The aim of this review is to summarize and evaluate our current understanding of the role of the gut microbiome, nutrition, and the ENS in stimulating maturation of the intestine, including reparative mechanisms that may impact age-dependent outcomes in severe equine colic cases.

## 2. The Anatomy of Small Intestinal Colic Injury in Foals

The intestine is lined with a single layer of epithelial cells responsible for preventing the entry of bacteria and their toxins while facilitating absorption of nutrients and water from the intestinal lumen [7,8]. This conflict in function between absorption and forming a barrier is regulated by the epithelial cells and interepithelial tight junctions. In the small intestine, the latter are relatively ‘leaky’ to enhance absorption. This ‘leakiness’ can be modulated directly by dietary nutrients. For example, glucose transport results in further opening of tight junctions to increase absorption [10]. However, this physiologic ‘leakiness’ can become pathologic during disease processes such as ischemic injury associated with surgical colic lesions resulting from loss of tight junction function and, ultimately, loss of epithelial cells in the form of epithelial sloughing.

The regions of the intestinal tract that are most susceptible to ischemic injury are predominantly based on the anatomical characteristics of the equine gastrointestinal tract. Specifically, in the small intestine, the duodenum is suspended by a short mesenteric attachment and is therefore immobile, whereas the jejunum and ileum are progressively more mobile due to an elongating mesenteric attachment from proximal to distal and are thus more susceptible to strangulating lesions with associated ischemic injury [11].

The small intestinal mucosa obtains oxygenated blood from the celiac and cranial mesenteric arteries, which are supported by the mesentery and branch off into a complex network of arcades arranged in parallel as the vasculature advances toward the intestinal wall and into the mucosa [12]. Due to its distance from systemic circulation, the villus tip is relatively hypoxic under normal conditions, and is the most susceptible to ischemic injury during strangulating obstruction [13,14,15]. Depending on the severity of the occlusion to blood flow and the length of time that the epithelium is hypoxic, the epithelium may slough, beginning at the villus tip within just fifteen minutes of ischemia, and progress down the villi toward the crypts, eventually sloughing the crypt base cells following four or more hours of ischemia [15]. This loss of epithelium compromises the barrier between the intestinal luminal bacteria and the intestinal capillaries and will lead to sepsis [16,17,18]. The pathophysiology of ischemic injury and damage inflicted to the organ varies with lesion type and influences the way cases are managed surgically. Perfusion of the tissue must be restored in order to save the organ and prevent death of the patient, but reperfusion may exacerbate the extent of the existing ischemic damage. Reperfusion injury, resulting from the detrimental effects of oxidative enzymes such as xanthine oxidase and secondary neutrophil migration, has been studied extensively and has been previously reviewed in detail [12,19,20,21].

Clinically studying age-related disparities in equine intestinal repair and survival following SISO is challenging because euthanasia decisions in equine hospitals can often be driven financially rather than medically and because the lesions are variable between age groups. Both foals and adults experience small intestinal volvulus, which can occur as hemorrhagic strangulating obstruction, as the veins collapse first due to their thinner vessel walls and lower blood pressure relative to arteries [11]. Loss of venous drainage with temporary maintenance of arterial blood supply results in a large increase in interstitial pressure that disrupts the intestinal architecture, but as the intestinal interstitial pressure increases, or as the rotation of the volvulus continues, the arteries will also be occluded, and blood flow into and out of the area will cease [11]. Less commonly, complete ischemia can occur in volvulus cases when the ‘twist’ of the strangulation provides sufficient pressure to occlude both the arteries and veins simultaneously. Adult horses, particularly aged geldings, can develop pedunculated lipomas that loop around and strangulate the intestine, a lesion not seen in foals. Some lesions may be similar enough for comparison between age groups. For example, a foal with volvulus secondary to an ascarid impaction can be compared to an adult with volvulus secondary to an impaction of feed material, though care must be taken to consider that intestinal parasites may elicit an immune response that leads to increased inflammation and postoperative complications not seen in the adult lesion [11]. Nonetheless, severe strangulating colic can impact a horse of any age and cannot always be prevented; therefore, understanding how colic varies in foals and adult horses helps to guide clinical management before and following surgical correction of the lesion. Further, understanding age-related differences in the mucosal microenvironment will help clinicians refine treatment strategies during recovery to optimize barrier repair, stop bacterial translocation, and prevent fatal complications.

## 3. Efficient Mucosal Repair Limits Morbidity from Colic

Following injury, efficient epithelial barrier restitution and repair prevents translocation of bacteria and their products and the resultant detrimental clinical consequences. In the small intestine of mature animals, acute repair begins with restitution, which consists of rapid villous contraction and epithelial migration, and concludes with intercellular tight junction closure. This early phase of repair is completed within the span of just a few hours. Villus contraction begins with the prostaglandin-mediated activation of the ENS, which initiates immediate contraction of myofibroblasts below the basal lamina and prolonged contraction of smooth muscle cells within the villus core [16]. Simultaneously, epithelial cells begin to shrink in height and spread out, crawling to completely cover the denuded surface. Epithelial cell lamellipodia lead this migration through reorganization and cycling of the actin and myosin components of the cytoskeleton while maintaining contact with the basement membrane through integrin attachments [16,22,23,24]. Once the wound beds are covered with the existing, newly restituted epithelium, tight junctions must then be restored to close their paracellular spaces and return epithelial cellular polarity (arrangement of apical versus basolateral domains of the cell). Tight junction proteins are reinserted into the cell membrane through the function of endosomal recycling [21,25]. This step is critical to complete barrier repair as interlinked tight junction proteins prevent bacterial toxins from continuing to cross the barrier, reducing exacerbation of sepsis and associated complications [26]. Following restoration of the barrier, a return to normal intestinal architecture is driven by the proliferation of intestinal stem cells located in the intestinal crypts. Intestinal stem cells proliferate within the crypts, pushing neighboring cells up the crypt villus axis until normal intestinal architecture is regained. Intestinal stem cells are highly active under physiological conditions as the intestinal epithelium completely turns over under normal conditions every 5–7 days, depending upon the species. During the subacute phase of mucosal repair, intestinal stem cells become even more proliferative to return the mucosa to its normal architecture over the days to weeks following injury. Currently researched therapeutics, such as pro-repair molecules and stem cell therapies, aim to maximize the efficiency of both early and late repair mechanisms [27,28].

### 3.1. Evidence of Age-Dependent Barrier Repair: A Comparative Pig Model

Because clinically recording differences in barrier repair is challenging, studies often rely on animal models of gastrointestinal barrier disruption and repair, particularly in neonatal and early life gastrointestinal research [29,30]. The pig has proven to be a powerful large animal model of gastrointestinal physiology for translation to humans and large animals as it can overcome some limitations of mouse and other rodent models [31]. The pig has been shown to be a viable model for equine gastrointestinal research, where similarities in both anatomy and physiology are of use experimentally [20,32]. The horse and the pig share similar microvascular anatomy within the small intestine, where a single, eccentrically located arteriole and similar branching of the venules within the villus will exhibit comparable epithelial sloughing patterns in response to ischemic injury [12,14,20]. Anatomically, though, there are species differences in the ENS; specifically, differences in the number of layers in the myenteric and submucosal plexuses. These differences have been characterized in the mouse and human and, more recently, in the pig, and it is reasonable to assume that there may be such anatomical differences between the pig and the horse. Though there are minor anatomical differences, functional differences between mammalian species have yet to be fully studied.

Physiologically, the neonatal pig is a good model for ischemia/reperfusion injury in the foal as both species lack intestinal xanthine dehydrogenase, an enzyme involved in the physiologic breakdown of hypoxanthine to xanthine and then to uric acid, at birth, indicating that reperfusion injury may not be as much of a concern in neonates of either species as xanthine dehydrogenase is converted to xanthine oxidase in hypoxic conditions, which produces superoxide during reperfusion [20]. Further, the pig has been established as a powerful model of early life stress, with important implications for the development of lifelong gastrointestinal disease [29,33,34]. One pig study identified a complete defect in ex vivo restitution of the intestinal barrier after ischemic injury in two-week-old piglet jejunum after 30 to 120 min of ischemia [3]. This defect, however, could be rescued by application of a homogenate of ischemic-injured mucosa from weanling-aged animals [3]. This indicates some aspect of the mucosa is not present or is immature in neonates that is apparently present and functional in weanling animals, and research is ongoing to determine the mechanisms responsible for this disparity which is of particular importance to foals suffering from severe colic.

### 3.2. A Complex Mucosal Microenvironment Regulates Epithelial Repair

Barrier repair is regulated intensively by complex and coordinated subepithelial cell signaling events. There are many cellular components of the subepithelial microenvironment to consider when investigating age-related differences in intestinal repair. The small intestine is a complex organ comprised of several layers, the outermost being the serosa surrounding the longitudinal and circular muscle layers. The smooth muscle layers facilitate motility, driving segmentation, churning, and forward transit of digesta, while ensuring adequate contact time between the luminal contents and the epithelium to increase absorption of digested nutrients. Between the two layers of smooth muscle is the myenteric plexus of the ENS, a network of intrinsic primary afferent neurons, interneurons, excitatory and inhibitory neurons, and enteric glial cells that regulate smooth muscle activity among many other emerging functions [9,35].

Internal to the smooth muscle layers is the submucosa, a region of loose connective tissue containing vasculature, lymphatic vessels, and the submucosal plexus of the ENS, another network of neurons and enteric glial cells (Figure 1). This region also houses a diverse population of immune cells, mesenchymal cells, and endothelial cells. Intestinal mesenchymal cells consist of fibroblasts, myofibroblasts, pericytes, mesenchymal stem cells, smooth muscle cells, interstitial cells of Cajal, and fibrocytes [36]. These cells function to provide mechanical support, epithelial homeostasis, stem cell maintenance, immune regulation, extracellular matrix maintenance, angiogenesis, and vascular function regulation [36]. Studies have identified a therapeutic effect when inflamed and damaged intestine is treated with mesenchymal stem cells in vitro [37,38,39]. Endothelial cells form the barrier between the intravascular elements and the submucosal microenvironment and are responsible for the induction of inflammation and recruitment of leukocytes through the release of several proinflammatory cytokines and colony-stimulating factors [40]. The innermost layer is the intestinal mucosa, containing subepithelial capillaries and lymphatic vessels, neuronal projections, and yet another network of enteric glial cells with projections extending close enough to directly contact the single-cell thick epithelial barrier. This epithelial population includes a carefully coordinated and organized population of absorptive and secretory enterocytes, enteroendocrine cells, goblet cells, tuft cells, and intestinal stem cells [41].

On the opposite side of the epithelial barrier, the small intestinal lumen contains ingesta, mucus, and two microbial populations, one that is suspended within the ingesta and another that is adherent to the mucus layer between the luminal contents and the mucosa. Within these two populations, microbes can be commensal, symbiotic, or pathogenic, and the intestinal microbiota is increasingly implicated broadly in health and disease [42,43]. Mucus is continually secreted by goblet cells in the small intestine and provides both a chemical and physical barrier to the potentially harmful bacteria in the intestinal lumen [44]. Loss of this mucus layer, through ischemic injury for example, increases intestinal permeability and the risk of patient sepsis [45]. Dietary components found in the intestinal lumen also function to stimulate mucus secretion and absorption while other nutrients function to decrease barrier activity. For example, glucose causes intracellular tight junctions to open the paracellular space, allowing nutrients and water to cross the barrier more easily [10]. These components of the luminal microenvironment are equally important to consider when investigating age-dependent mucosal repair following ischemic intestinal injury.

## 4. Factors Influencing Postnatal Maturation of Intestinal Repair Mechanisms

### 4.1. Enteric Nervous System-Microbiome Interactions

The ENS orchestrates many functions of the gut, including maintenance of peristalsis and motility to transport food and waste, regulation of secretory reflexes, sensation of luminal nutrients, regulation and maintenance of a microbial population, maintenance of the epithelial barrier, and regulation of inflammation in response to pathogens, toxins, and dietary allergens [46]. Given the importance of the ENS to the individual in maintaining proper barrier function and absorption of nutrients, it is easy to understand that improper development of the ENS has been implicated in many gastrointestinal diseases and disorders in many species.

Studies have shown ablation of enteric neurons and glial cells causes severe intestinal inflammation [47], and recent literature points to defects in ENS development as a major contributing or causal factor in several developmental gastrointestinal diseases and functional disorders. Some of these disorders include ileocolonic aganglionosis, which is Hirschsprung’s disease in humans and lethal white foal syndrome in horses, pseudo-obstruction syndrome, and a subset of irritable bowel syndrome patients may have defects in ENS development [47,48]. Given the emerging importance of the enteric glia in developmental diseases in horses, ongoing work to examine the postnatal maturation of the enteric glial network in the horse by visualizing and quantifying the enteric glial network at discrete postnatal timepoints in foals will be informative (Figure 2).

While there are many factors that could drive or influence postnatal maturation and modulation of intestinal repair mechanisms, the ENS is very likely a large contributor. Most intestinal anatomy and physiological functions are considered fully developed at birth, with the exception of the ENS, and particularly the enteric glial network, which are both implicated in coordinating critical signaling within the subepithelial microenvironment. Relatively close proximity of ENS components to the epithelium is required to allow for cell–cell signaling during these processes. Enteric glial cells are restricted to the myenteric and submucosal plexuses at the time of birth in rodent models, and recent literature reports that migration of enteric glial cells into the lamina propria is thought to be driven, at least in part, by intestinal microbial colonization in early postnatal life. Foals commonly experience mild diarrhea early in life, referred to as “foal heat diarrhea”, which is likely caused by the ongoing colonization of the gut and subsequent hypersecretion of the intestine, highlighting another link between the ENS and microbiome [49]. Further, microbiota have been shown to directly impact the intestinal epithelium through direct signaling with various apical toll-like receptors [50]. Toll-like receptor 4 has been specifically shown to decrease barrier function in intestinal epithelial cell lines when stimulated with lipopolysaccharide, a bacterial component, through upregulation of myosin light chain kinase, which subsequently dysregulates intercellular tight junction proteins [51]. LPS activation of toll-like receptor 4 has also been shown to increase pyroptosis in enteric neurons, further solidifying the direct functional link between bacteria and the ENS [52].

Newly colonizing commensal bacteria are beneficial to the neonate, as they cooperatively extract and synthesize nutrients and metabolites through the fermentation of luminal contents that would not be available otherwise and competitively exclude pathogenic species of bacteria [53,54]. Butyrate, a microbial metabolite, has been specifically shown to modulate the postnatal development of the enteric glial network [55]. The action of butyrate on the enteric glial network further supports the idea that postnatal development of the ENS is driven by diet-influenced changes in microbial populations and subsequent microbial metabolites. Studies have shown that a microbial population is required for development of the glial network postnatally as germ-free mice failed to fully develop glial projections from the submucosal plexus up to the epithelial surface, but this network was established following colonization of germ-free mice with microbiota from conventionally raised individuals [56,57]. Signaling between microbial metabolites and enteric glia may be implicated in driving these postnatal changes, but the mechanisms of these interactions are not yet well understood [55]. Because the ENS appears to develop postnatally in response to luminal signals in the form of microbes or their metabolites, modulating microbial populations earlier in life may enhance the speed at which the ENS matures following birth and subsequently increase favorable outcomes in young patients with severe intestinal injury.

### 4.2. Microbiota

Intestinal dysbiosis in humans has been linked to several gastrointestinal disorders such as inflammatory bowel disease and celiac disease, to neurodegenerative diseases such as Parkinson’s disease, and to cardiovascular disease [43,58,59,60,61]. In horses, the literature is less extensive, but this is a growing area of research and similar links have been made between gastrointestinal disorders, microbiota, and the ENS [62,63,64]. Alterations in intestinal microbial populations have been identified in equine colitis, postpartum colic, and chronic laminitis [63]. Similar to humans, recent literature has identified the mare as an important part of the foal’s microbiome development [65]. Quercia et al. noted similarities in the bacterial components found in mare amniotic fluid and foal meconium, with some overlap in the mare feces and foal meconium microbial populations, and foal and mare fecal microbial populations begin to converge within the first week of life, attributed to foal coprophagy [65,66].

Prebiotics and probiotics have been used with variable results in horses [64]. Thus, fecal microbiota transplants have become a subject of renewed interest and research after the clinical success of treating *Clostridium difficile* infections in human patients, where 15 of 16 patients treated with fecal microbial transplantation were cured, as compared to 4 of 13 patients who received the standard vancomycin treatment [63,67]. *Clostridium difficile* produces two main toxins which destroy the intestinal barrier through loss of epithelial cytoskeletal components and disruption of intercellular tight junction proteins. Further, *Clostridium difficile* activates secretomotor neurons in the ENS, initiating severe secretory diarrhea [68]. Given the success of fecal microbial transplantation in human patients, targeting the equine enteric microbiota in a similar way could be an important tool in modulating intestinal repair mechanisms and in preventing or treating gastrointestinal injury in young horses.

### 4.3. Nutrition

Nutritional factors are known to directly modulate permeability of the intestinal barrier, as digested nutrients must be recognized and allowed to cross the epithelium during absorption [69]. Specifically, acetaldehyde and fatty acids like EPA, DHA, and lauric acid have been shown to increase permeability in Caco-2 cells in vitro [69]. Alternatively, probiotic strains *S. thermophilus*, *L. acidophilus*, *L. rhamnosus*, and *L. plantarum*, and vitamins A and D have all been shown to reduce permeability in Caco-2 cell cultures [69]. In addition to modulating the intestinal barrier, nutritional components can also modulate enteric neuron neurotransmission and plasticity [70]. Particularly, colostrum has been shown to be important to piglet nitric oxide synthase, which is involved in vasoactive intestinal peptide-dependent adaption and survival of enteric neurons [70]. The antibodies in colostrum may further be important to modulating the response to colonizing enteric bacteria [70].

While nutritional factors have been shown to directly modulate barrier function, indigestible dietary components, termed prebiotics, are fermented by commensal bacteria into bioactive metabolites which may also impact intestinal function. Prebiotics have been shown to decrease recovery time following *Salmonella* infection in suckling neonatal pigs, decrease recovery time of intestinal architecture following malnourishment, and decrease barrier restitution time following intestinal injury [71]. Of particular importance, are short chain fatty acids (SCFAs) including acetate, propionate, and butyrate, which are produced through the bacterial fermentation of indigestible dietary fibers. SCFAs provide a significant source of energy to the horse, and fermentation of the SCFA precursors maintains the sensitive microbial populations in the equine colon and cecum. Dietary prebiotics may also aid in competitive exclusion of pathogenic bacteria and in stabilization of a normal microbial population by fueling desired bacterial species. Speculatively, prebiotic supplementation in early development might drive earlier maturation of the foal intestinal microbiome, which may, in turn, drive earlier migration of the enteric glia into the submucosal space, increasing the efficiency of intestinal repair mechanisms and possibly survival following surgical correction of SISO.

### 4.4. Early Life Stress

In addition to microbiota and nutrition having an influence on postnatal development of the enteric glial network, early life events and stressors may also play an important role. The stress of weaning has been shown to play a role in intestinal disease, and it is possible that weaning stress and other early life stressors might further encourage the maturation of intestinal repair mechanisms in humans and other mammalian species [29,33,34,72]. Normal weaning stress has been shown to create marked decreases in intestinal barrier function and an increase in corticotrophin-releasing factor in the pig [73], while the further increased stress of early weaning can induce chronic functional diarrhea [33]. Early weaning stress has also been noted in horses, as the accepted ages for weaning in managed and feral horse populations vary greatly. The accepted weaning age for managed horses is between 4 and 6 months of age, and weaning typically consists of an abrupt and permanent separation of the foal from the mare [74]. This is in contrast to the weaning process observed in feral horse populations, where weaning only occurs when the dam is due to foal again, around 11 or 12 months into the foal’s life, and weaning occurs gradually over time with maintenance of the foal’s relationship to the mare [74]. This study noted a decrease in salivary cortisol, the release of which is stimulated by corticotrophin-releasing factor, and a decrease in stress behaviors such as vocalization, active locomotion, startle response to sudden stimuli, and the time taken to fit a halter in foals that were progressively separated from their dam when compared to foals that were suddenly and definitively separated [74]. In humans, early life adversity has been linked to adult functional gastrointestinal disorders, and higher levels of psychological distress in the form of anxiety and depression in early childhood increases the likelihood of developing irritable bowel syndrome as an adult [33]. A porcine model of early weaning stress has been able to recapitulate the effects on human infants in an experimental setting, and may be applicable to weaning stress in horses as well [33]. In a surgical ischemia model, it has also been noted that juvenile pigs subjected to acute transport stress on the day of intestinal ischemic injury exhibit increased intestinal injury but accelerated epithelial restitution when compared to pigs that experienced more protracted psychosocial stress related to prolonged changes in housing environment prior to injury [75]. These findings all indicate that stress responses early in life are likely to have major impacts on intestinal repair in foal colic cases.

## 5. Conclusions

Colic in foals cannot always be prevented, so understanding the causes, symptoms, and patterns of injury and healing can improve interventions available to improve outcomes for young patients. Presently, the neonatal defect in restitution identified in the pig model in a prior study is thought to be due to the relatively immature ENS in the postnatal period. The mechanisms of ENS maturation are not yet well characterized, and research to identify contributions of dietary inputs, microbiota, the ENS, and stress will be critical. In equine patients, this enteric glial network maturation may vary from other species as the foal microbiome may be greatly impacted by coprophagy beginning in the first week of life and natural creep feeding of small amounts of grain, hay, and grass within the first two months of life, well before true weaning. Understanding how these processes drive maturation of intestinal repair mechanisms in foals can inform the development of new management strategies, preventative measures, and clinical interventions to mitigate the severity of injury and enhance repair responses to reduce morbidity and mortality rates among foals experiencing severe colic.

## Figures and Tables

**Figure 1 animals-11-03337-f001:**
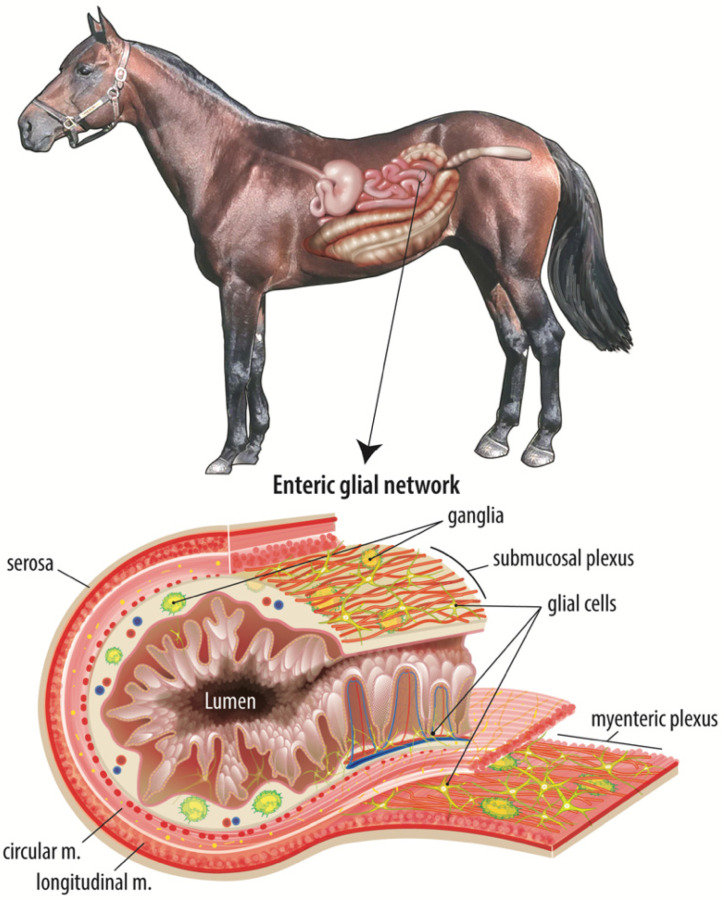
The adult equine enteric glial network. The intestinal mucosal microenvironment is home to several cell types that all work in a complex, coordinated manner to maintain the epithelial barrier and restore the barrier in response to intestinal injury. Of these cell types, the enteric glial network is thought to act as an intermediary between the enteric neurons and luminal signals, such as nutrients or microbial metabolites. This figure illustrates the complexity and expanse of the enteric glial network, including directly adjacent to the epithelium.

**Figure 2 animals-11-03337-f002:**
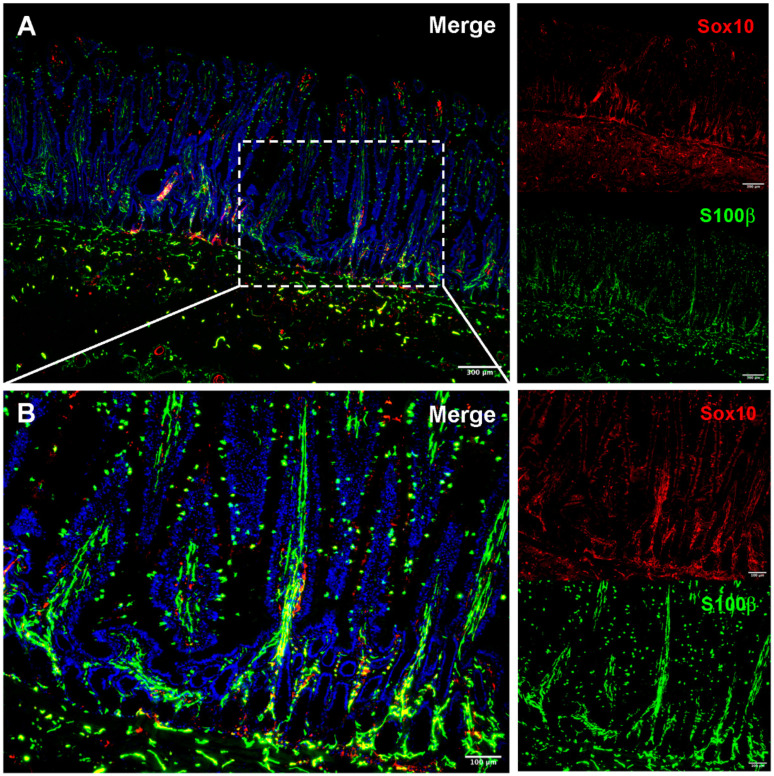
Immunofluorescence visualization of the adult equine enteric glial network. Immunofluorescence allows thin-slice imaging of the enteric glial network in many species, including the horse. Small intestinal sections of tissue reveal relatively high S100β (green) expression throughout the intestinal wall when compared to sparse expression of Sox10 (red) (**A**). The intricacy of the enteric glial network can be seen in the mucosa when viewed at higher magnification (**B**). Ongoing work to image the equine small intestine and colon in three-dimensions will quantify the differences in the glial network between foals and adult horses and inform future study of enteric glia in early life gastrointestinal disease.

## Data Availability

No new data were created or analyzed in this review article. Data sharing is not applicable to this article.

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
