# Peer review of "Age-Dependent Intestinal Repair: Implications for Foals with Severe Colic"

_animals, 2021, doi:10.3390/ani11123337_

Round 1

Reviewer 1 Report

Although very interesting and generally well designed, the reviewed manuscript contains some issues that needs to be clarified before the publication.

  1. The authors did not mention that there is an anatomical differences in the structure of the ENS between the pig and horse. The submucous plexus of the porcine small intestine is further divided into outer and inner submucous plexus, whereas such division is not present in the horse.
  2. The authors seem to have completely forgotten about the plasticity of the intestinal neurons. Meanwhile, many factors (including nutritional ones) evoke changes in chemical code of enteric neurons. It seems than in young animals one of the most important may be feeding with colostrum, which has been proved to influence the neurotransmitter content in ENS (see Woliński J. et al. Effect of feeding colostrum versus exogenous immunoglobulin G on gastrointestinal structure and enteric nervous system in newborn pigs. J Anim Sci. 2012 Dec;90 Suppl 4:327-30). This topic and work must be included in the manuscript.
  3. The authors should also mention that there is a direct functional link between the gut microbiota and ENS. The expression of TLR4 has been detected in enteric neurons and a bacterial product LPS has been found to be neurotoxic to different populations of enteric neurons.

Author Response

Thank you for your thoughtful review and helpful feedback and suggestions. Please find below our responses to your comments:

Point 1: “The authors did not mention that there is an anatomical differences in the structure of the ENS between the pig and horse. The submucous plexus of the porcine small intestine is further divided into outer and inner submucous plexus, whereas such division is not present in the horse.”

Response: Thank you for pointing out this missing information. We have made edits to reflect this on lines 177-182 in the revised manuscript. However, we were not aware that someone has done this work in the horse. Can you share these references? We would be grateful.

Point 2: “The authors seem to have completely forgotten about the plasticity of the intestinal neurons. Meanwhile, many factors (including nutritional ones) evoke changes in chemical code of enteric neurons. It seems than in young animals one of the most important may be feeding with colostrum, which has been proved to influence the neurotransmitter content in ENS (see WoliÅ„ski J. et al. Effect of feeding colostrum versus exogenous immunoglobulin G on gastrointestinal structure and enteric nervous system in newborn pigs. J Anim Sci. 2012 Dec;90 Suppl 4:327-30). This topic and work must be included in the manuscript.”

Response: Thank you for pointing this out. We have included this reference with a brief description of the work on lines 369-374 in the revised manuscript.

Point 3: “The authors should also mention that there is a direct functional link between the gut microbiota and ENS. The expression of TLR4 has been detected in enteric neurons and a bacterial product LPS has been found to be neurotoxic to different populations of enteric neurons.”

Response: Thank you for this suggestion to expand on the functional link between the microbiota and the ENS. We have included this in the manuscript with appropriate citations on lines 302-209 in the revised manuscript.

Reviewer 2 Report

Comments to the Authors of manuscript number: animals-1471419 entitled “Age-dependent Intestinal Repair: Implications for Foals with Severe Colic”.

The authors have presented a review about intestinal repairing, which could be helpful in understanding of colic in horses. There is a few gaps, which should be explained. Some of them are listed below. Each point should be explained.

Moreover, if Authors have mentioned health problems in various species, may be it is worth to change the title. The journal of Animals is rather dedicated for animals only, thus each part concerning humans should be omitted.

  1. L 13 – in general simple summary is very good.
  2. L 13- one work?
  3. L 13- what is it? –“various intestinal systems?
  4. L 8 – is this review about the problem only in the United States? Or can concerns to the general problem?
  5. L 18 – is it really rapidly? E.g. after diarrhea the repairing is observed after 6 weeks.
  6. L 20- omit humans
  7. L 51 – omit humans, Animals is not about humans
  8. L 72- the reference should be added
  9. L 73- not our but common
  10. L 79-82 – reference should be added
  11. L 93-109 – very good part
  12. L e.g. 139 interepithelial tight junction – it is rather intercellular tight junction
  13. L 143 – can we speak about cells traveling? The epithelial sells are rising from the crypts.
  14. L 154 – where are located these stem cells?
  15. L 174 – it should be explained shortly for what serves xanthine dehydrogenase
  16. Why there is not presented that END includes both submucosal and intramuscular ganglion? Figure 1 is missing them.
  17. L 323- omit humans
  18. L 344- ethanol in horses
  19. L 384- 386- omit humans
  20. L 388 – omit humans

Author Response

Thank you very much for your thorough and thoughtful review. Please find our responses to your comments detailed here:

Point 1, 2, & 3 in reference to line 13: Thank you for the compliment regarding the simple summary. We realize that the wording was a little unclear, especially regarding the description of the collective work in the field. In order to clarify this, we changed the word “work” to the plural “works” to better encompass all of the research in the field. Further, we provided an example of the “various intestinal systems” to help give readers a preview of the systems that we detail in the abstract. We were attempting to further simplify the simple summary, and concede that the wording there was a bit vague.

Point 4 in regards to line 8: The reference we used is from the United States Department of Agriculture, but we recognize that this problem applies to the rest of the world as well, so we changed the wording to reflect that it is a worldwide problem in the equine industry.

Point 5 in reference to line 18: In this section of the text, and in most of the paper, we are referring to acute intestinal barrier repair, which we refer to as restitution which is independent of proliferation. Specifically, the remaining epithelial cells spread and crawl to cover the denuded area. We also describe  longer term repair, often referred to as remodeling, which is driven by intestinal stem cells in the crypts proliferating to replace the epithelial cells lost to injury. We hope this explanation helps to clarify this point.

Points 6, 7, 16, 18, and 19 requesting that omission of any human information: We can appreciate your perspective that Animals is geared toward non-human animal species. We respectfully prefer to include the human data referenced in our manuscript. Because an enormous proportion of very rigorous biomedical research is geared towards human health and because humans are also mammals, we feel that reference to human biology and biomedicine is necessary to encompass a complete review of the current literature surrounding this topic. Many important aspects of gastrointestinal tract physiology in mammals are very similar across species, including humans. While we acknowledge that horses and humans are not the same species, the wealth of data from human-focused studies are important for a comprehensive review of what is known about age-dependency of intestinal injury and repair.

Point 8 referring to line 72: Thank you for pointing out this missing citation; we have added an appropriate reference.

Point 9 referring to like 73: thank you for pointing out that this area can be clarified; we have adjusted the wording to reflect this.

Point 10 referring to lines 79-82: Thank you for pointing out this missing reference; appropriate citations have been added.

Point 11 referring to lines 93-109: Thank you for this compliment- it is greatly appreciated.

Point 12 referring to line 139: Thank you for pointing out this possibly confusing term. It has been changed to "intercellular tight junction" in the manuscript.

Point 13 referring to line 143: Thank you for pointing out that this needed to be clarified. Here, we weren’t speaking about stem cells differentiating to replace lost epithelium, but we were speaking about the very first steps in acute repair, termed restitution, when the remaining epithelial cells flatted and spread, crawling to cover the surface area that lost epithelial cells. We have changed the wording to clarify this point. 

Point 14 referring to line 154: Thank you for pointing this out. We have added that these stem cells reside in the crypt base.

Point 15 referring to line 174: Thank you for this suggestion. We have added in a very brief explanation of what xanthine dehydrogenase does.

Point 15, referring to figure 1: We apologize that this is unclear, but there are ganglia in both the submucosal plexus and the intramuscular plexus. In our figure, we refer to the intramuscular plexus as the Myenteric plexus, but you will see that the Myenteric plexus does have ganglia in our figure.

Point 17 line 344: Thank you for pointing this out. While ethanol is used as a neurolytic agent to induce joint fusion in horses with severe OA, we recognize that ethanol is unlikely to be introduced into the gut, and so have omitted this information from the manuscript.

Reviewer 3 Report

Excellent review. Congratulations to the authors.

Author Response

Thank you very much for your kind comments.

Round 2

Reviewer 1 Report

The manuscript has been corrected according my suggestions.